# COPD Patients with Asthma Features in Vietnam: Prevalence and Suitability for Personalized Medicine

**DOI:** 10.3390/jpm13060901

**Published:** 2023-05-26

**Authors:** Nguyen Van Tho, Thu Phuong Phan, Anh Tuan Dinh-Xuan, Quy Chau Ngo, Le Thi Tuyet Lan

**Affiliations:** 1Department of Tuberculosis and Lung Diseases, University of Medicine and Pharmacy at Ho Chi Minh City, Ho Chi Minh City, Vietnam; thonguyen0225@ump.edu.vn; 2Department of Pulmonary Functional Exploration, University Medical Center at Ho Chi Minh City, Ho Chi Minh City, Vietnam; 3Department of Internal Medicine, Ha Noi Medical University, Ha Noi City, Vietnam; thuphuongdr@gmail.com (T.P.P.); ngoquychaubmh@gmail.com (Q.C.N.); 4Respiratory Center, Bach Mai Hospital, Ha Noi City, Vietnam; 5AP-HP, Hôpital Cochin, Service de Physiologie-Explorations Fonctionnelles, Paris, France; anh-tuan.dinh-xuan@aphp.fr; 6Tam Anh General Hospital, Ha Noi City, Vietnam

**Keywords:** asthma, chronic obstructive pulmonary disease, COPD, overlap

## Abstract

COPD patients with asthma features usually benefit from inhaled corticosteroids (ICS)-containing regimens, but their burden and diagnostic criteria remain to be established. The aims of this study were to estimate the proportion of patients with asthma features among patients with physician-diagnosed COPD and to investigate differences in clinical characteristics and current medications between COPD patients with asthma features and patients with COPD alone. A cross-sectional study was conducted at two respiratory out-patient clinics at the University Medical Center in Ho Chi Minh City and Bach Mai Hospital in Ha Noi, Vietnam. COPD patients with asthma features were identified by attending physicians following the approach recommended by the GINA/GOLD joint committee. Of the 332 patients screened, 300 were enrolled in the study. The proportion of COPD patients with asthma features was 27.3% (95% confidence interval (95% CI) 22.6–32.6%). COPD patients with asthma features were younger, with higher FEV_1_ values, a greater proportion of positive bronchodilator reversibility tests, higher blood eosinophil count, and were more often treated with ICS/LABA (ICS/long-acting bronchodilator beta-2 agonist) than patients with COPD alone. The prevalence of COPD patients with asthma features is particularly high in Vietnam thus requiring appropriate action plans in clinical practice.

## 1. Introduction

Patients with COPD have heterogeneous clinical presentation, imaging characteristics, decline in lung function, response to therapy, and survival [1]. Approximately 15% to 20% of COPD patients present with features of asthma such as a history of atopy or allergic rhinitis, airway hyper-responsiveness, or laboratory evidence of eosinophilia in the blood or sputum [2,3]. These patients are considered to have overlapping asthma and COPD, commonly referred to as asthma–COPD overlap (ACO) or asthma + COPD [3,4] or COPD with asthma features [5]. COPD patients with asthma features represent a distinct phenotype of COPD and have worse health-related quality of life scores, more frequent and severe symptom exacerbations, and a greater incidence of acute respiratory events compared with patients with COPD alone [6,7]. Management of COPD patients with asthma features therefore requires more medical resources and expenditure due to significantly more frequent emergency room visits and hospital and intensive care unit admissions [8].

Identifying asthma features in a patient with COPD is the first step towards developing a personalized treatment plan [5,9]. COPD patients with asthma features usually benefit from inhaled corticosteroids (ICS), while patients with COPD alone may not [3,5]. However, diagnosing COPD patients with asthma features is a challenge in clinical practice [10,11]. Since the first publication of the GINA/GOLD (Global Initiative for Asthma/Global Initiative for Chronic Obstructive Lung Disease) joint committee recommendations for diagnosing ACO in 2015 [12], several alternative diagnostic criteria have been proposed by different experts or professional societies [13,14,15,16]. Physicians face the challenge of choosing which diagnostic criteria to apply because they do not know which ones are clinically relevant for their individual healthcare setting [1,17]. The prevalence of COPD patients with asthma features varies widely among different populations because it largely depends on the chosen diagnostic criteria [18,19,20]. Therefore, the burden of COPD patients with asthma features remains to be established [17,21]. The aims of this study were to estimate the proportion of patients with asthma features among patients with physician-diagnosed COPD in Vietnam and to investigate differences in the clinical characteristics and current medications between COPD patients with asthma features and patients with COPD alone.

## 2. Patients and Methods

### 2.1. Study Design

This was a cross-sectional, non-interventional study performed at the respiratory clinics of the University Medical Center in Ho Chi Minh City and Bach Mai Hospital in Ha Noi City, Vietnam. Patients were enrolled via consecutive sampling from September 2016 to June 2017. All patients with COPD who visited the hospital out-patient clinics were screened for potential enrolment. During the screening process, the investigators recorded patient information, explained the study to the patient, and screened the patient against the study inclusion and exclusion criteria. After adequate explanation of the study, the investigator obtained written informed consent from all the patients who met the inclusion criteria and had no conflict with the exclusion criteria. This study did not modify the way patients with COPD were evaluated and treated by attending physicians in daily practice.

### 2.2. Patients

All patients with physician-diagnosed COPD at both clinics were recruited from September 2016 to June 2017. Physicians diagnosed COPD using the GOLD criteria, which include chronic respiratory symptoms, exposure to noxious particles or gases, and a post-bronchodilator FEV_1_/FVC (ratio of forced expiratory volume in one second (FEV_1_) to forced vital capacity (FVC)) of less than 70% [5]. Patients with COPD were included if they fulfilled all the following criteria: age >40 years old; COPD diagnosed for at least one year; seen at an out-patient clinic; and in a stable state. Patients were excluded from the study if they met any one of the following criteria: acute exacerbations of COPD as defined by GOLD (any worsening of a patient’s respiratory symptoms that is beyond normal day-to-day variation and requires a change in medications) within the past 6 weeks; respiratory diseases that present similar symptoms to chronic airway diseases such as bronchiectasis, pulmonary tuberculosis, endobronchial tuberculosis, and lung cancer or a history of these diseases based on the attending physician’s judgement; currently diagnosed with pneumonia or acute bronchitis; or currently randomized in other clinical studies.

### 2.3. Data Collection

Investigators interviewed patients and collected the following clinical characteristics: demographics, smoking history, family history (asthma, allergies, atopy), comorbidities, number of exacerbations in the previous year, respiratory symptoms, severity of dyspnea based on the modified Medical Research Council (mMRC) dyspnea scale, and quality of life based on COPD Assessment Test (CAT) score [22]. Investigators retrospectively collected patients’ spirometric results from medical records to show evidence of COPD, including the most recent, the best, and the worst measurements over the past three years. Patients performed standardized spirometry at both clinics using the KoKo spirometer (nSpire Health Inc., Longmont, CO, USA) before and after inhaling 400 μg of salbutamol (Ventolin, GlaxoSmithKline, Middlesex, UK) at the time of initial COPD diagnosis. Patients performed spirometry without a bronchodilator test at all follow-up visits. All spirometric maneuvers met American Thoracic Society/European Respiratory Society standards [23]. Spirometric parameters were expressed as absolute values and percentage of predicted values based on the reference equations of NHANES III, with a correction factor of 0.88 for Asians [24]. Other test results that were collected if they were available in the medical record included blood eosinophil count and chest X-ray. The investigators also collected information about the current medications that were prescribed to each patient by the attending physicians.

### 2.4. Diagnosing COPD Patients with Asthma Features

The diagnosis of COPD patients with asthma features was at the discretion of the attending physicians. Those physicians applied the step-wise approach recommended by the GINA/GOLD joint committee criteria in this study [12]. In short, COPD patients with asthma features were diagnosed when patients with COPD had at least 3 features of asthma: symptoms that started before the age of 40; symptoms that worsen at night or in the early morning; symptoms that are triggered by exercise, emotional change (including laughter), or dust/allergens; symptoms that vary either seasonally, year to year, or over time; symptoms that do not worsen over time; symptoms that improve spontaneously; symptoms that improve significantly over weeks with medications containing ICS; and spirometric results that show very positive bronchodilator reversibility—FEV_1_ change ≥15% and ≥400 mL after inhaling 400 µg salbuterol [12].

For experimental purposes, the investigators retrospectively identified COPD patients with asthma features based on diagnostic criteria proposed by other experts or professional societies as follows: “physician-diagnosed asthma” criteria, if the COPD patients had ever been diagnosed as having asthma before 40 years old by doctors [7]; “modified Spanish expert consensus” criteria, if the COPD patients had either “physician-diagnosed asthma” criteria or the criteria of very positive bronchodilator reversibility (FEV_1_ change ≥15% and ≥400 mL after inhaling 400 µg salbuterol) [14]; “FEV_1_ variation over time” criteria, if the COPD patients had either “physician-diagnosed asthma” criteria or FEV_1_ difference between the best and the worst measurements ≥15% and ≥400 mL [25].

### 2.5. Statistical Analysis

The sample size was calculated assuming the proportion of COPD patients with asthma features was 20%. With a confidence level of 95%, the minimum sample size was estimated to be 246. Categorical variables were described as numbers and percentages. Continuous variables were described as mean and standard deviation if their distributions were normal or as median and interquartile range if their distributions were skewed. Comparisons of continuous variables between COPD patients with asthma features and patients with COPD alone were performed using Student’s *t*-test if their distribution was normal or the Kruskal–Wallis test if their distribution was skewed. Comparisons of categorical variables between COPD patients with asthma features and patients with COPD alone were performed using the chi-square or Fisher’s exact test, where appropriate. Statistical analyses were performed using JMP 9.0.2 (SAS Institute Inc., Cary, NC, USA). A *p*-value of less than 0.05 was considered statistically significant.

## 3. Results

### 3.1. The Proportion of COPD Patients with Asthma Features

Of the 332 COPD patients who were screened, 300 were enrolled in the study. Of those enrolled, the number or proportion of COPD patients with asthma features identified by the attending physicians using the “GINA/GOLD joint committee” criteria was 82 or 27.3% (95% confidence interval (95% CI) 22.6–32.6%). Table 1 shows the differences in the frequency of features that favored asthma or COPD between COPD patients with asthma features and patients with COPD alone collected by the attending physicians. All features favoring asthma were significantly more common in patients with asthma features except for the recording of reversible airflow limitation (*p* = 0.828). In contrast, features favoring COPD were significantly less common in patients with asthma features except for the recording of persistent airflow limitation (*p* = 0.941), response to short-acting bronchodilators (*p* = 0.050), and signs of hyperinflation on a chest X-ray (*p* = 0.225).

The numbers and proportions of COPD patients with asthma features retrospectively identified by the investigators using the “physician-diagnosed asthma”, the “modified Spanish expert consensus”, and the “FEV_1_ variation over time” criteria were 34 (11.3%; 95% CI 8.2–15.4%), 38 (12.7%; 95% CI 9.4–16.9%), and 34 (11.3%; 95% CI 8.2–15.4%), respectively. Figure 1 shows the Venn diagram of the number of COPD patients with asthma features among the 300 patients with COPD by different diagnostic criteria. The number of COPD patients with asthma features identified using the “FEV_1_ variation over time” criteria was the same as the number identified using the “physician-diagnosed asthma” criteria (*n* = 34). Most COPD patients with asthma features identified using the “modified Spanish expert consensus” criteria overlapped with those identified using the “physician-diagnosed asthma” criteria (*n* = 34), except for four patients. By using the “GINA/GOLD joint committee” approach, the attending physicians identified most (30 out of 38) COPD patients with asthma features identified by the three other diagnostic criteria and 52 more patients not identified by those three diagnostic criteria.

### 3.2. Characteristics of COPD Patients with Asthma Features

The following analyses were focused on COPD patients with asthma features identified by using the “GINA/GOLD joint committee” criteria as these helped the attending physicians to identify the most COPD patients with asthma features among the four diagnostic criteria. Table 2 shows that most demographic characteristics between COPD patients with asthma features and patients with COPD alone were not significantly different. However, COPD patients with asthma features were younger (*p* = 0.021), had symptoms onset at a younger age, were more likely to be women (*p* < 0.001), and were less likely to be smokers (including current smokers and ex-smokers) (*p* < 0.001) than patients with COPD alone.

Table 3 shows that differences in COPD exacerbations, hospitalizations in the previous year, CAT, and mMRC were not statistically significant between COPD patients with asthma features and patients with COPD alone. However, blood eosinophil count or its percentage change was significantly greater in COPD patients with asthma features than in patients with COPD alone (*p* = 0.016 or *p* = 0.004, respectively). Table 3 also shows that the proportion of eosinophilia was higher in COPD patients with asthma features than in patients with COPD alone for both the cut-offs of 300/µL and 3% (*p* = 0.038 and *p* = 0.007, respectively).

Table 4 shows differences in the spirometric measurements of COPD patients with asthma features and patients with COPD alone. At the most recent measurement, FEV_1_ was greater in COPD patients with asthma features than in patients with COPD alone (*p* < 0.001). When performing the bronchodilator reversibility test, the proportions of positive (FEV_1_ increase ≥200 mL and ≥12%) or very positive bronchodilator reversibility tests (FEV_1_ increase ≥400 mL and ≥15%) were significantly higher in COPD patients with asthma features than in patients with COPD alone (*p* < 0.001 or *p* = 0.017, respectively). Among the 77 COPD patients with asthma features displaying initial bronchodilator reversibility testing, the proportions of positive reversibility testing or very positive reversibility testing did not significantly differ between male and female patients (28.6% vs. 2.6%, *p* = 0.324, or 6.5% vs. 0.0%, *p* = 0.582, respectively) or between smokers and non-smokers (23.4% vs. 7.8%, *p* = 0.340, or 3.9% vs. 2.6%, *p* = 0.657, respectively). When evaluating FEV_1_ changes over time, the proportion of patients with an FEV_1_ increase of ≥400 mL and ≥15% was significantly higher in COPD patients with asthma features than in patients with COPD alone (*p* = 0.035).

Table 5 shows that COPD patients with asthma features were more likely to be prescribed ICS/LABA (long-acting bronchodilator beta-2 agonist) (*p* = 0.012) or a leukotriene receptor antagonist (*p* < 0.001) but less likely to be prescribed LABA alone (*p* = 0.031) or theophylline (*p* = 0.001) than patients with COPD alone. ICS/LABA was the most common medication prescribed for COPD patients with asthma features (89.0%) and patients with COPD alone (76.6%).

## 4. Discussion

This study shows that, in Vietnam, the proportion of COPD patients with asthma features identified by attending physicians using the “GINA/GOLD joint committee” criteria was 27.3%. This proportion was higher than that identified retrospectively by the investigators using other diagnostic criteria such as the “modified Spanish expert consensus” (12.7%), the “physician-diagnosed asthma” (11.3%), and the “FEV_1_ variation over time” (11.3%) criteria. This study also showed that COPD patients with asthma features were younger, more likely to be female, less likely to be smokers, had a better FEV_1_, had a higher proportion of a positive or very positive bronchodilator reversibility testing, had a higher proportion of blood eosinophilia, and were more likely to be prescribed ICS/LABA than patients with COPD alone.

The findings of this study are in accordance with those of previous studies. Different proportions of COPD patients with asthma features have been reported from different populations when different diagnostic criteria were utilized [1,26,27,28]. The proportion of ACO among 568 patients with chronic respiratory symptoms at Pham Ngoc Thach Hospital, Vietnam, was 32% when a proposed diagnostic approach was applied [16]. The proportion of ACO among 522 patients with COPD in Canada varied from 3.8% to 31.0% when each of seven different definitions was used [19]. Another study involving 1008 COPD patients in Japan reported a prevalence of ACO of 16.6% when the step-wise approach recommended by the GINA/GOLD joint committee was applied [25]. The proportion of ACO among 1015 patients with a diagnosis of COPD in a primary care setting using the United Kingdom’s Optimum Patient Care Research Database was 20.5% [29]. A recent study in Spain showed that the proportion of ACO among 603 patients with COPD was 27.4% [30]. Of the 396 COPD patients who had the data necessary to diagnose ACO using the Japanese Respiratory Society criteria [13], the proportion of patients with ACO was 25.5% [31]. Findings from the two later studies and our own suggest that the more features that are used to identify ACO or asthma features, the higher proportion of patients with COPD will be diagnosed as COPD patients with asthma features.

Findings of this study imply that the “GINA/GOLD joint committee” criteria might identify more COPD patients with asthma features than any other diagnostic criteria. This finding is plausible given the fact that the “GINA/GOLD joint committee” criteria include more asthma features than other diagnostic criteria [11]. Because asthma features are treatable traits, COPD patients with asthma features can benefit from appropriate treatments that include ICS, respond better to currently available treatments, and have a better prognosis if treated properly [32]. In clinical practice, physicians should identify as many patients with asthma features among COPD patients as possible because treatments containing ICS help to reduce exacerbations or hospitalizations [33] and improve the FEV_1_ [34], and treating asthma without ICS is associated with safety concerns [3]. The “GINA/GOLD joint committee” criteria used in our study include clinical features and tests that are routinely collected, which reflects the clinical practice in out-patient clinics for patients with asthma and COPD. In a busy daily practice, clinicians tend to use clinical features and spirometric results rather than other tests to identify patients with asthma features because other tests are not always available, and they are either not convenient or time consuming to perform or both.

This work estimated how often features favoring asthma or COPD were collected or recorded to identify COPD patients with asthma features. The following asthma features were collected more frequently: symptoms that vary over time; symptoms that worsen during the night or early morning; symptoms that are triggered by exercise, emotional change, or exposure to dust/allergens; and symptoms that improve over weeks with ICS-containing treatment. This study also showed that records of reversible or persistent airflow limitation on spirometry or signs of hyperinflation on a chest X-ray were of limited value in identifying asthma features among patients with COPD.

Only a few patients with COPD (2.5%) fulfilled the criteria of an FEV_1_ increase ≥400 mL and ≥15% at their first spirometric measurement. This finding agrees with that of the ECLIPSE study, in which only 5% of patients with COPD had an FEV_1_ that increased by at least 400 mL after inhaling 400 µg salbutamol [35]. A very positive bronchodilator reversibility test at one visit may therefore add little value to identifying COPD patients with asthma features in clinical practice. However, 18.8% patients with COPD met the criteria of very positive bronchodilator reversibility when comparing their best and worst measurements over 3 years of follow up. This means that follow up is necessary to thoroughly diagnose most COPD patients with asthma features. This study also showed that both blood eosinophil cut-offs (300/µL and 3%) were helpful for differentiating COPD patients with asthma features from patients with COPD alone (*p* = 0.038 and *p* = 0.007). A recent study showed that COPD patients with asthma features and eosinophilia had a poor health-related quality of life [36].

Most COPD patients with asthma features in this study received ICS/LABA according to GINA recommendations [3] or experts’ consensus [11]. The attending doctors at University Medical Center in Ho Chi Minh City and Bach Mai Hospital in Ha Noi City, Vietnam, both tertiary and teaching hospitals, manage patients following international guidelines. The finding that 18.8% patients with COPD had a FEV_1_ that improved by ≥400 mL and ≥15% over 3 years of follow up simply reflects the better efficacy of currently prescribed medications in COPD patients with asthma features as compared to in patients with COPD alone (30.6% vs. 13.6%). A recent systematic review showed that ICS/bronchodilator-containing arms had a reduction of 7.3 mL/year (95% CI, 4.1 to 10.5) in the rate of FEV_1_ decline compared with the placebo arms [37]. Our results are therefore consistent with the idea that current pharmacological therapy may modify disease progression in patients with COPD. 

Some patients with COPD were treated with LTRA due to comorbidity of allergic rhinitis and/or asthma features in accordance with GINA recommendations or experts’ consensus [3]. Some patients also received theophylline prescribed by attending doctors due to persisting symptoms despite LABA/LAMA or ICS/LABA/LAMA treatments. The prescription of LTRA or theophylline among COPD patients with asthma features implies that attending doctors have tried different available medications that might provide some beneficial effects in both asthma and COPD. This might also reflect the paucity of pharmacological means to treat COPD patients with asthma features. This pattern of medications prescription in Vietnam may change when there is more evidence about the cost effectiveness of certain medications in COPD patients with asthma features.

This study had several limitations. First, the identification of asthma features using the approach recommended by the “GINA/GOLD joint committee” depends on the physician’s clinical experience. Therefore, generalization of the proportion of COPD patients with asthma features from this study to other clinical settings should be made with caution. Second, there is a concern that some COPD patients, such as female patients, non-smokers, or patients meeting the criteria of positive bronchodilator reversibility testing at the first spirometry, could have been wrongly labelled as COPD with asthma features when these patients were actually late-onset asthma. However, we believe that these patients were patients with COPD, not patients with late-onset asthma, because only COPD patients with the following features were included: COPD had been diagnosed for at least one year by attending doctors who followed GOLD recommendations; and had a post-bronchodilator FEV_1_/FVC < 70%. In addition, not only patients with asthma, but a proportion of patients with COPD met the criteria of positive bronchodilator reversibility testing [38]. The fact that we did not find a statistical difference in positive bronchodilator reversibility testing either between female and male patients or between smokers and non-smokers suggests that late-onset asthma was not associated with gender or smoking status in our study. Third, because this was a cross-sectional study, we do not know whether COPD patients with asthma features diagnosed using the “GINA/GOLD joint committee” criteria would have a different response to current treatments or a different prognosis. A longitudinal study should be performed to answer this research question. Fourth, few patients with COPD in this study underwent a blood eosinophil count, so we do not know which underlying mechanisms COPD patients with asthma features diagnosed using the “GINA/GOLD joint committee” criteria were experiencing Further studies with other biomarkers such as fractional exhaled nitric oxide, sputum eosinophil counts, and respiratory microbiome labelling are needed [39].

## 5. Conclusions

The prevalence of COPD patients with asthma features is relatively high in Vietnam, accounting for more than one quarter of patients with COPD. They have different characteristics and use different medications than patients with COPD alone. In clinical practice, COPD patients with asthma features need to be thoroughly identified by taking a detailed medical history and utilizing pulmonary function tests or other biomarkers. They should be individually prescribed currently available medications based on evidence generalized from randomized control trials and observational studies.

## Figures and Tables

**Figure 1 jpm-13-00901-f001:**
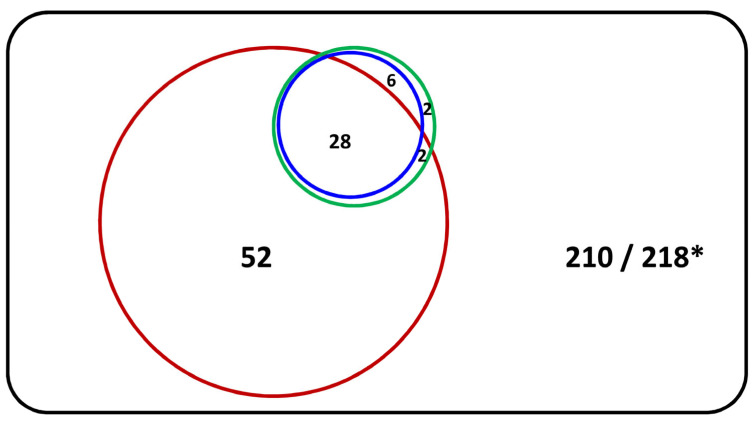
Venn diagram of the number of COPD patients with asthma features by different diagnostic criteria. Notes: red circle, number of COPD patients with asthma features by the “GINA/GOLD joint committee” criteria; blue circle, number of COPD patients with asthma features by either the “physician-diagnosed asthma” or the “FEV_1_ variation over time” criteria; green circle, number of COPD patients with asthma features by the “modified Spanish expert consensus”; and black rectangle, number of patients with COPD alone by the combination of the 4 above-mentioned diagnostic criteria (* *n* = 210) or by the “GINA/GOLD joint committee” criteria (* *n* = 218).

**Table 1 jpm-13-00901-t001:** Frequency of features favoring asthma or COPD according to the GINA/GOLD joint committee.

Features	COPD with Asthma Features * (*n* = 82)	COPD Alone * (*n* = 218)	*p*-Value
**Features that favor asthma**			
Symptoms vary over time	43 (52.4%)	40 (18.4%)	**<0.001**
Symptoms vary seasonally	19 (23.2%)	9 (4.1%)	**<0.001**
Symptoms worsen during the night or early morning	42 (51.2%)	18 (8.3%)	**<0.001**
Symptoms are triggered by exercise, emotional change, or exposure to dust/allergens	42 (51.2%)	23 (10.6%)	**<0.001**
Symptoms do not worsen over time	17 (20.7%)	3 (1.4%)	**<0.001**
Symptoms improve spontaneously	16 (19.5%)	15 (6.9%)	**0.002**
Symptoms improve over weeks with ICS treatment	37 (45.2%)	15 (6.9%)	**<0.001**
Record of reversible airflow limitation	4 (4.9%)	12 (5.5%)	0.828
Allergic rhinitis	13 (15.9%)	3 (1.4%)	**<0.001**
Physician-diagnosed asthma	28 (34.1%)	6 (2.8%)	**<0.001**
Family history of asthma	9 (11.0%)	7 (3.2%)	**0.016**
**Features that favor COPD**			
Symptoms onset after 40 years old	68 (89.5%)	207 (97.6%)	**0.006**
Symptoms persist despite treatment	33 (40.2%)	146 (67.0%)	**<0.001**
Good/bad days but symptoms are observed on a daily basis	22 (26.8%)	130 (59.6%)	**<0.001**
Chronic cough and sputum precede onset of dyspnea and are unrelated to triggers	40 (48.8%)	159 (72.9%)	**<0.001**
Symptoms slowly worsen over time	52 (63.4%)	181 (83.0%)	**<0.001**
Rapid-acting bronchodilator treatment provides only limited relief	34 (41.5%)	118 (54.1%)	0.050
Record of persistent airflow limitation	80 (97.6%)	213 (97.7%)	0.941
Abnormal lung function between symptoms	(*n* = 52) 52 (100.0%)	(*n* = 117) 111 (94.9%)	**0.034**
Hyperinflation on chest X-ray	(*n* = 52) 21 (40.4%)	(*n* = 73) 45 (31.0%)	0.225

Notes: Data are presented as number (%). * Identified by attending physicians using the “GINA/GOLD joint committee” criteria. Bold text in the *p*-value column represents a statistically significant difference. Abbreviations: GINA, Global Initiative for Asthma; GOLD, Global Initiative for COPD; ICS, inhaled corticosteroids.

**Table 2 jpm-13-00901-t002:** Demographic characteristics of 300 patients with COPD.

Characteristics	Total (*n* = 300)	COPD with Asthma Features * (*n* = 82)	COPD Alone * (*n* = 218)	*p*-Value
Age (years)	65.0 ± 9.7	62.7 ± 10.6	65.8 ± 9.2	**0.021**
Age at symptoms onset (years)	60.5 ± 12.2	56.0 ± 15.8	62.1 ± 10.2	**0.004**
Female	24 (8.0%)	14 (17.1%)	10 (4.6%)	**<0.001**
Weight (kg)	55.4 ± 9.7	55.9 ± 8.8	55.2 ± 10.0	0.572
Height (cm)	160.1 ± 6.7	159.7 ± 6.8	160.2 ± 6.6	0.585
BMI (kg/m^2^)	21.6 ± 3.6	21.9 ± 3.4	21.5 ± 3.6	0.345
Current smoker	53 (17.7%)	15 (18.3%)	38 (17.4%)	**<0.001**
Ex-smoker	202 (67.3%)	40 (48.8%)	162 (74.3%)	
Non-smoker	45 (15.0%)	27 (32.9%)	18 (8.3%)	
Pack-years	(*n* = 237) 33.7 ± 17.9	(*n* = 53) 34.0 ± 18.7	(*n* = 184) 33.6 ± 17.7	0.884
Comorbidities				
Hypertension	56 (18.7%)	16 (19.5%)	40 (18.4%)	0.818
Allergic rhinitis	16 (5.5%)	13 (15.9%)	3 (1.4%)	**<0.001**
GERD	16 (5.5%)	4 (4.9%)	12 (5.5%)	0.828
Ischemic heart disease	14 (4.7%)	4 (4.9%)	10 (4.6%)	0.916
Diabetes mellitus	10 (3.3%)	2 (2.4%)	8 (3.7%)	0.585

Notes: Data are presented as mean ± standard deviation or number (%) or median (interquartile range), where appropriate. * Identified by attending physicians using the “GINA/GOLD joint committee” criteria. Bold text in the *p*-value column represents a statistically significant difference. Abbreviations: COPD, chronic obstructive pulmonary disease; BMI, body mass index; GERD: gastro-esophageal reflux disease.

**Table 3 jpm-13-00901-t003:** Clinical and blood eosinophilic characteristics of 300 patients with COPD.

Characteristics	Total (*n* = 300)	COPD with Asthma Features * (*n* = 82)	COPD Alone * (*n* = 218)	*p*-Value
≥1 exacerbation in the previous year	90 (30.0%)	22 (26.8%)	68 (31.2%)	0.459
≥2 exacerbations in the previous year	45 (15.0%)	8 (9.8%)	37 (17.0%)	0.106
≥1 hospitalization in the previous year	45 (15.0%)	11 (13.4%)	34 (15.6%)	0.634
CAT	(*n* = 181) 16.0 ± 6.3	(*n* = 45) 14.8 ± 5.6	(*n* = 136) 16.4 ± 6.5	0.111
CAT ≥ 10	151 (83.4%)	37 (82.2%)	114 (83.8%)	0.804
mMRC	(*n* = 179) 2 (2; 3)	(*n* = 54) 2 (2; 3)	(*n* = 125) 2 (2; 3)	0.091
mMRC ≥ 2	143 (79.9%)	42 (77.8%)	101 (80.8%)	0.646
Blood eosinophil count (per µL)	(*n* = 36) 190 (102.5; 297.5)	(*n* = 10) 310 (162.5; 650)	(*n* = 26) 145 (100; 215)	**0.016**
Blood eosinophil percent	2.3% (1.2%; 3.8%)	4.4% (2.2%; 6.7%)	1.9% (1.0%; 2.8%)	**0.004**
Blood eosinophil ≥ 300/ µL	9 (25.0%)	5 (50.0%)	4 (15.4%)	**0.038**
Blood eosinophil ≥ 3%	12 (33.3%)	7 (70.0%)	5 (19.2%)	**0.007**

Notes: Data are presented as mean ± standard deviation or number (%) or median (interquartile range), where appropriate. * Identified by attending physicians using the “GINA/GOLD joint committee” criteria. Bold text in the *p*-value column represents a statistically significant difference. Abbreviations: CAT, COPD assessment test; mMRC, modified Medical Research Council dyspnea scale.

**Table 4 jpm-13-00901-t004:** Spirometric characteristics of 300 patients with COPD.

Characteristics	Total	COPD with Asthma Features *	COPD Alone *	*p*-Value
Latest spirometric measurement				
	(*n* = 201)	(*n* = 54)	(*n* = 147)	
% post-BD FEV_1_	54.6 ± 18.1	61.7 ± 16.1	52.0 ± 18.2	**<0.001**
% post-BD FVC	74.2 ± 16.8	80.8 ± 14.4	71.8 ± 17.0	**<0.001**
Post-BD FEV_1_/FVC	53.6 ± 11.6	57.4 ± 8.4	52.2 ± 12.3	**<0.001**
Bronchodilator reversibility at the first spirometric measurement				
	(*n* = 284)	(*n* = 77)	(*n* = 207)	
FEV_1_ change (mL)	80 (20; 170)	120 (50; 215)	70 (20; 150)	**0.001**
%FEV_1_ change (%)	7.9 (2.1; 15.7)	11.8 (2.7; 21.6)	7.0 (1.7; 14.2)	**0.014**
FEV_1_ increase ≥200 mL and ≥12%	45 (15.8%)	24 (31.2%)	21 (10.1%)	**<0.001**
FEV_1_ increase ≥400 mL and ≥15%	7 (2.5%)	5 (6.5%)	2 (1.0%)	**0.017**
FEV_1_ change between the best and worst measurements during 3 years of follow up				
	(*n* = 117)	(*n* = 36)	(*n* = 81)	
Pre-BD FEV_1_ change (mL)	240 (85; 420)	300 (170; 600)	200 (75; 355)	**0.012**
% pre-BD FEV_1_ change (%)	8 (3; 17)	10 (4; 24)	7.0 (3; 15.5)	0.298
FEV_1_ increase ≥200 mL and ≥12%	45 (38.5%)	16 (44.4%)	29 (35.8%)	0.377
FEV_1_ increase ≥400 mL and ≥15%	22 (18.8%)	11 (30.6%)	11 (13.6%)	**0.035**

Notes: Data are presented as mean ± standard deviation or number (%) or median (interquartile range), where appropriate. * Identified by attending physicians using the “GINA/GOLD joint committee” criteria. Bold text in the *p*-value column represents a statistically significant difference. Abbreviations: COPD, chronic obstructive pulmonary disease; BD, bronchodilator; FEV1, forced expiratory volume in one second; FVC, forced vital capacity.

**Table 5 jpm-13-00901-t005:** Maintenance medications prescribed by attending physicians for 300 patients with COPD.

Medications	Total (*n* = 300)	COPD with Asthma Features * (*n* = 82)	COPD Alone * (*n* = 218)	*p*-Value
LABA ^†^	42 (14.0%)	6 (7.3%)	36 (16.5%)	**0.031**
LAMA	104 (34.7%)	24 (29.3%)	80 (36.7%)	0.224
LABA/ICS ^‡^	240 (80.0%)	73 (89.0%)	167 (76.6%)	**0.012**
LABA/ICS + LAMA ^§^	70 (23.3%)	20 (24.4%)	50 (22.9%)	0.791
LTRA	50 (16.7%)	25 (30.5%)	25 (11.5%)	**<0.001**
Theophylline	92 (30.7%)	14 (17.1%)	78 (35.8%)	**0.001**

Notes: Data are presented as number (%). * Identified by attending physicians using the “GINA/GOLD joint committee” criteria. ^†^ Including inhaled and oral long-acting beta-2 agonists; ^‡^ 66.4% (158/238) used salmeterol/fluticasone propionate delivered via a pressurized metered-dose inhaler, and 33.6% used formoterol/budesonide delivered via a turbuhaler; ^§^ LABA/ICS and LAMA used in two separate inhalers. Bold text in the *p*-value column represents a statistically significant difference. Abbreviations: COPD, chronic obstructive pulmonary disease; ICS, inhaled corticosteroids; LABA, long-acting beta-2 agonist; LAMA, long-acting muscarinic antagonist; LTRA, leukotriene receptor antagonist.

## Data Availability

All data not published within this article will be made available by request from any qualified investigator.

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
