# Peer review of "COPD Patients with Asthma Features in Vietnam: Prevalence and Suitability for Personalized Medicine"

_jpm, 2023, doi:10.3390/jpm13060901_

Round 1

Reviewer 1 Report

I am very impressed with the work, the quality of data preparation and presentation. Although the topic of ACO seems to be heavily explored in recent years, prospective, observational studies describing this group of patients are still needed.

I have no significant comments, I recommend the publication of this manuscript.

Author Response

Thank you for your kind comments on the manuscript.

Reviewer 2 Report

A well written article that is easy to read with very little grammatical errors. 

A few major comments : 

A very big portion (24 patients, 31.2%) of patients with a positive reversibility test were labelled to have COPD with asthma features. Another 6.5% had a very positive reversibility test in this group of patients. What proportion of these patients were female and never smokers and what were the age of onset of the symptoms? My concern is "are these truly COPD patients with asthma features?" or could some of them have been wrongly diagnosed as COPD when what they truly have is just asthma? 

The next major comment is how can the authors explain the increase in reversibility testing in the COPD alone group of patients when followed up over 3 years? 

In the group of patients receiving LRTA, did the physician felt that the diagnosis was asthma rather than COPD as the treatment is not in accordance to the GOLD guidelines? 

In Figure 1, there are 2 numbers (210 and 218) in the black rectangle. What does the 210 represent? 

The discussion paragraphs seems bulky and can be breached down better into smaller paragraphs. They should also address the above comments I've given above in the discussion section. 

Reviewer 3 Report

I read the manuscript with interest. The following comments should be addressed:

- At the end of the introduction (aims) the authors have to state that they estimated the proportion in a specific city/country.

- Methods: mention the date of the study in the study design rather than the patients section (or in both).

- Line 187 and discussion: "more likely to be non-smokers (P<0.001" is a misleading, especially that (slightly) higher current smoker exists in the ACO group, There are no p values for the other smoking categories.

- Table 1: More clear description of comorbidities is needed; I mean the percentages of those with comorbidities (not only the n) and the percentage of the overall.

- Introduction and discussion: some references need more updates including the prevalence and the diagnostic and treatments criteria. Consider the following very recent (2023) references, including the recent review published by the same Journal (JPM):

Alsayed AR, Abu-Samak MS, Alkhatib M. Asthma-COPD Overlap in Clinical Practice (ACO_CP 2023): Toward Precision Medicine. Journal of Personalized Medicine. 2023 Apr 18;13(4):677.

*  Chu HT, Nguyen TC, Godin I, Michel O. A Proposal to Differentiate ACO, Asthma and COPD in Vietnam. Journal of Personalized Medicine. 2023 Jan;13(1):78.

*  Alsayed AR, Abed A, Jarrar YB, Alshammari F, Alshammari B, Basheti IA, Zihlif M. Alteration of the Respiratory Microbiome in Hospitalized Patients with Asthma–COPD Overlap during and after an Exacerbation. Journal of Clinical Medicine. 2023 Mar 8;12(6):2118.

- I suggest adding the appendix table to the original manuscript.

- The conclusion needs more clear details and recommendations

Some minor grammar errors

Round 2

Reviewer 2 Report

The authors have added this into the results : 

 We have added the age at symptoms onset in the Table 2. 

We have modified the Results section (page 7, line 225-230) specifying that the proportions of positive reversibility were not significantly higher in female or in non-smokers than in male or in smokers. “Among 77 patients with asthma features displaying initial bronchodilator reversibility, the proportions of positive reversibility tests or very positive reversibility tests did not significantly differ between male and female (28.6% vs 2.6%, p=0.324 or 6.5% vs 0.0%, p=0.582, respectively) or between smokers and non-smokers (23.4% vs 7.8%, p=0.340 or 3.9% vs 2.6%, p=0.657, respectively).” 

However, this statement is very confusing and does nothing to address the concern that the 14 female patients (17.1%) and 27 non-smoker patients (32.9%) as well as 24 patients (31.2%) who fulfilled the positive bronchodilator reversibility test at first initial spirometry could have been wrongly mislabelled as COPD with asthma features when all these patients are having are actually just late-onset asthma. 

Reviewer 3 Report

The manuscript is now much improved and appreciated efforts. However, I suggest adding "in Vietnam" at the end of the title or following "features" word because the current title is more suitable for reviews.

Author Response

Thank you for your suggestion.

We have added "in Vietnam" following "features" word in the Title.